# Strawberry *FaSnRK1α* Regulates Anaerobic Respiratory Metabolism under Waterlogging

**DOI:** 10.3390/ijms23094914

**Published:** 2022-04-28

**Authors:** Jingjing Luo, Wenying Yu, Yuansong Xiao, Yafei Zhang, Futian Peng

**Affiliations:** State Key Laboratory of Crop Biology, College of Horticulture Science and Engineering, Shandong Agricultural University, Tai’an 271018, China; luo.jingj@163.com (J.L.); ying26231@163.com (W.Y.); ysxiao@sdau.edu.cn (Y.X.)

**Keywords:** strawberry, *FaSnRK1α*, roots, anaerobic respiratory, *ERFVIIs*, waterlogging

## Abstract

Sucrose nonfermenting-1-related protein kinase 1 (SnRK1) is a central integrator of plant stress and energy starvation signalling pathways. We found that the *FaSnRK1α*-overexpression (OE) roots had a higher respiratory rate and tolerance to waterlogging than the *FaSnRK1α*-RNAi roots, suggesting that *FaSnRK1α* plays a positive role in the regulation of anaerobic respiration under waterlogging. *FaSnRK1α* upregulated the activity of anaerobic respiration-related enzymes including hexokinase (HK), phosphofructokinase (PFK), pyruvate kinase (PK), pyruvate decarboxylase (PDC), alcohol dehydrogenase (ADH) and lactate dehydrogenase (LDH). *FaSnRK1α* also enhanced the ability to quench reactive oxygen species (ROS) by increasing antioxidant enzyme activities. We sequenced the transcriptomes of the roots of both wild-type (WT) and *FaSnRK1α*-RNAi plants, and the differentially expressed genes (DEGs) were clearly enriched in the defence response, response to biotic stimuli, and cellular carbohydrate metabolic process. In addition, 42 genes involved in glycolysis and 30 genes involved in pyruvate metabolism were significantly regulated in *FaSnRK1α*-RNAi roots. We analysed the transcript levels of two anoxia-related genes and three *ERFVIIs*, and the results showed that *FaADH1*, *FaPDC1*, *FaHRE2* and *FaRAP2.12* were upregulated in response to *FaSnRK1α*, indicating that *FaSnRK1α* may be involved in the ethylene signalling pathway to improve waterlogging tolerance. In conclusion, *FaSnRK1α* increases the expression of *ERFVIIs* and further activates anoxia response genes, thereby enhancing anaerobic respiration metabolism in response to low-oxygen conditions during waterlogging.

## 1. Introduction

Oxygen is one of the necessary factors for plants to maintain normal physiological metabolism and to complete normal growth and development processes. However, poor drainage and irrigation and flood disasters often lead to excessive waterlogging, resulting in partial or total submergence and hypoxic stress within the plant rhizosphere. Under aerobic conditions, the root system carries out normal respiration and plays roles in the absorption, synthesis, and transport of compounds and other functions. Insufficient oxygen supply in the rhizosphere can disrupt physiological and biochemical metabolism and affect plant growth and development [1].

The response of plants to hypoxia stress can be divided into three stages. First, plants under hypoxia stress quickly produce signal transduction components. Then, the signalling molecules induce the expression of a large number of stress response genes, which regulate the plant fermentation pathway for energy supply and the antioxidant defence system to quench reactive oxygen species (ROS). Finally, plants form aerenchyma, adventitious roots and other morphological structures improve plant stress resistance [2,3,4].

Research on plant responses to flooding stress-related signalling molecules has mainly focused on ethylene. Soil flooding increases the ethylene synthesis rate [5] and reduces the rate of ethylene diffusion from the roots to the soil [6], and ethylene accumulates rapidly in the roots. The accumulation of ethylene in turn can induce the formation of plant aerenchyma [7] and adventitious roots [8].

The group VII APETALA 2/ETHYLENE-RESPONSE FACTOR (AP2/ERF) transcription factors (ERFVIIs) are key mediators of flooding and hypoxia stress responses in rice and *Arabidopsis* [9,10]. ERFVIIs regulate metabolic changes and developmental reprogramming under low-oxygen conditions [11,12,13]. *Arabidopsis* has five ERFVIIs, namely, RAP2.2, RAP2.3, RAP2.12, HRE1 and HRE2 [13,14,15,16]. During submergence, RAP2.12, RAP2.2 and RAP2.3 are stabilized and accumulate in the nucleus, affecting the transcription of stress response genes [16]. However, the *rap2.12-2 rap2.3-1* double mutant is hypersensitive to both submergence and osmotic stress. HRE1 and HRE2 play partially redundant roles in low-oxygen signalling, thus improving the tolerance to stress by enhancing anoxia-related gene expression and ethanolic fermentation. Through AtERF73/HRE1, ethylene regulates metabolic adaptation to low-oxygen stress in Arabidopsis roots [15,17]. The *hre1 hre2* mutant showed a significantly lower tolerance of anoxia. Conversely, *HRE1-* but not *HRE2*-overexpressing seedlings show enhanced anoxia tolerance [14].

Energy metabolism and antioxidant defence systems are the keys to enduring short-term flooding damage in plants. Flooding stress results in plant tissues being in a hypoxic or anaerobic state, and aerobic respiration and oxidative phosphorylation are blocked. The reduced plant energy supply is alleviated through glycolysis and fermentation processes. Under waterlogging stress, the activity of alcohol fermentation enzymes, including alcohol dehydrogenase (ADH) and pyruvate decarboxylase (PDC), increases. In Arabidopsis, *ADH1* and *PDC1* were shown to be upregulated in response to submergence [18]. Arabidopsis *ADH1* is regulated by ethylene [19] and *RAP2.2* [20].

Sucrose nonfermenting-1-related kinase 1 (SnRK1) is the central regulator of energy homeostasis in plants and is known to trigger a vast array of transcriptional and metabolic reprogramming in response to declining energy levels [21,22]. SnRK1 is an essential kinase induced by autophagy under various stress conditions in Arabidopsis. Moreover, SnRK1 activity is required for flooding resistance in Arabidopsis [18]. Kudahettige et al. [23] found that overexpression of the rice *SnRK1* gene can promote starch degradation and increase the energy supply in plants under waterlogging treatment. In rice, SnRK1A plays a key role in stimulating seed germination and seedling growth under normal and anaerobic (flooding) conditions [24,25,26]. Two hypoxia-inducible SnRK1A-interacting negative regulators (SKIN1/2) inhibit starch-to-sugar hydrolysis and delay germination and seedling growth under submergence [27].

Strawberry is a kind of perennial berry fruit tree with a shallow root system. During the propagation of strawberry stolon, some factors such as flooding, unfavourable irrigation and tight soil texture cause the occurrence of hypoxia around the strawberry root-zone. SnRK1 plays an important role in the submergence-triggered signalling pathway [28]. In this study, we reported that strawberry *FaSnRK1α*, which encodes a catalytic subunit of the SnRK1 protein kinase, can regulate the expression of the *HRE2* and *RAP2.13* genes, affect the expression of the downstream genes *PDC1* and *ADH1*, and therefore enhance the anaerobic respiration metabolism of plants roots under waterlogging conditions. Through studying on the effect of waterlogging on the root respiration and exploring the adaptive mechanism of strawberry to hypoxic stress, it can provide a theoretical basis for solving the problem of soil hypoxic stress. The results of our research provide new insights into the mechanisms by which *FaSnRK1α* promotes anaerobic respiration of plant roots.

## 2. Results

### 2.1. FaSnRK1α Regulates Root Activity and Respiratory Rate under Waterlogging

The roots of strawberry seedlings were infected with *A. rhizogenes* strain K599 harbouring the TRV2-*FaSnRK1α* (*FaSnRK1α*-RNA interference (RNAi)) and pCAMBIA1302-*FaSnRK1α* (*FaSnRK1α* overexpression) vectors, after which the plants were grown in substrate. Since the vectors contained the GFP gene, using an in vivo imaging system, we detected gene expression in the TRV-*FaSnRK1α* (RNAi) and pCAMBIA1302-*FaSnRK1α* (OE) transgenic roots one month after infection (Figure 1A,C). GFP signals were detected in the roots of the TRV-*FaSnRK1α* (RNAi) and pCAMBIA1302-*FaSnRK1α* (OE) strawberry plants but not in those of the controls, indicating that the gene was successfully transferred into the strawberry roots by *A. rhizogenes* strain K599.

To validate the inhibition of the *FaSnRK1α* gene at the molecular level, real-time PCR analysis was performed. The results showed that the *FaSnRK1α* transcripts were markedly downregulated in the roots of TRV-*FaSnRK1α* compared to those of the roots transformed with the TRV empty control; among them, the inhibition efficiency of *FaSnRK1α* in the TRV-*FaSnRK1α*-2 OE roots was the highest, reaching 48.02% (Figure 1B). Therefore, TRV-*FaSnRK1α*-2 was selected to silence the *FaSnRK1α* gene in subsequent experiments. The expression level of *FaSnRK1α* gene was up-regulated by 2.3-fold in the pCAMBIA1302-*FaSnRK1α* OE roots compared to that in the pCAMBIA1302 empty control roots (Figure 1D).

To investigate the effects of *FaSnRK1α* on root physiological characteristics, we measured the root activity and respiration rate in the roots of *FaSnRK1α* (OE) and silenced (RNAi) plants. As shown in Figure 2, under normal conditions (0 d), there was no significant difference in root activity between the OE or RNAi plants and their corresponding controls. However, the increases and decreases in root activity in the roots of the OE and RNAi plants were much more evident (a 21.65% increase and 30.07% decrease) than those of OE-C and RNAi-C plants after 6 d of waterlogging treatment (Figure 2A). Furthermore, the root respiration rate of the OE plants was significantly higher than that of the OE-C plants, with a 10.38–22.81% increase at 3, 6 and 9 d after waterlogging. In contrast, the silencing of *FaSnRK1α* resulted in a significant decrease (10.17–21.21%) in the respiration rate in the roots of those plants compared with the WT and RNAi-C plants under waterlogging (Figure 2B). Overall, these results indicate that *FaSnRK1α* overexpression (OE) or silencing (RNAi) leads to an increase or decrease, respectively, in root activity and respiration rate under waterlogging.

### 2.2. FaSnRK1α Regulates Anaerobic Respiration Metabolism under Waterlogging

Under waterlogging conditions, aerobic respiration is inhibited and plants undergo anaerobic respiration as a temporary adaptive response to hypoxic or anaerobic stress. In this process, the production of ATP, which is required for energy metabolism, mainly depends on the glycolysis pathway. As shown in Figure 3A–C, the activities of three key enzymes (HK, PFK and PK) in the glycolysis pathway showed an increasing trend for a short period of time, while the activities were lower than those under normal conditions (0 d) after a longer waterlogging period. This is because glycolysis is a common pathway of aerobic respiration and anaerobic respiration, and in the late stage of waterlogging, accompanied by exacerbated hypoxic stress and decreasing aerobic respiration, glycolysis-related enzyme activities decreased. Overexpression of *FaSnRK1α* increased the enzyme activities of HK, PFK and PK in the OE roots, which were 23.98%, 21.94% and 19.72% higher than those in the OE-C roots, respectively. In contrast, the activities of HK, PFK and PK in the *FaSnRK1α*-RNAi roots were 12.21% 21.90% and 17.29% lower than those in RNAi-C roots, respectively. In conclusion, *FaSnRK1α* increases the activity of key enzymes in glycolysis, promoting glycolysis and resulting in the production of increased amounts of ATP to adapt to the energy shortage during waterlogging.

In the process of glycolysis, the dehydrogenation of glyceraldehyde 3-phosphate requires the coenzyme NAD^+^_,_ which can be regenerated by ethanol and lactate fermentation in the absence of oxygen. Therefore, we further analysed the activities of enzymes related to alcohol and lactate fermentation. As shown in Figure 3D,E, the activities of PDC, ADH and lactate dehydrogenase (LDH) increased sharply after waterlogging treatment, indicating that anaerobic respiration metabolism was enhanced. Compared with that of the OE-C roots, the activity of PDC, ADH and LDH in the OE roots increased by 18.25%, 14.45% and 40.29%, respectively, while the enzyme activity in the RNAi roots decreased by 20.58%, 25.79% and 22.76%, respectively, compared with that of the RNAi-C roots. Overall, these results indicate that *FaSnRK1α* increased the activity of related enzymes during ethanol fermentation and lactic acid fermentation, thereby enhancing anaerobic respiration metabolism under waterlogging.

### 2.3. FaSnRK1α Regulates ROS Accumulation and Antioxidant Enzyme Activities under Waterlogging

To further investigate whether *FaSnRK1α* can enhance root resistance to hypoxic stress, the correlations between *FaSnRK1α* and reactive oxygen metabolism in plants and the differences in antioxidant enzyme activities as well as ROS content between the WT and transgenic plants were studied. After 3–12 d of waterlogging, the overexpression of *FaSnRK1α* (OE) resulted in significant decreases (37.16% and 19.38% on average) in the O^2−^ and H_2_O_2_ contents compared with those of the OE-C plants, while the silencing of *FaSnRK1α* (RNAi) resulted in increases (20.51% and 7.93% on average) in the O^2−^ and H_2_O_2_ contents (Figure 4A,B).

An increase in ROS content can lead to oxidative stress, and by scavenging free radicals, the antioxidant enzyme system in cells can maintain the relative balance of reactive oxygen metabolism. Therefore, we measured the antioxidant enzyme activities in the roots of *FaSnRK1α*-OE and *FaSnRK1α*-RNAi plants under waterlogging. As shown in Figure 4C–E, the increases in superoxide dismutase (SOD), peroxidase (POD) and catalase (CAT) activities in the roots of *FaSnRK1α*-OE plants were much more evident than they were in the OE-C plants, with 24.99%, 25.85% and 26.79% increases, respectively, on average, after waterlogging treatment. However, compared with those of the RNAi-C plants, the roots of the *FaSnRK1α*-RNAi plants exhibited lower SOD, POD and CAT activities under waterlogging. Taken together, these results indicated that *FaSnRK1α* may play a role in antioxidant systems to protect plants during waterlogging stress.

### 2.4. Transcriptomic Analysis of Roots of WT and FaSnRK1α-RNAi Plants

To further evaluate the molecular mechanism through which *FaSnRK1α* regulates responses to waterlogging, we performed a transcriptome analysis on the roots of *FaSnRK1α*-RNAi (fasnrk1a) and WT (control) under normal conditions (BioProject accession number: PRJNA826700). Three biological replicates were tested using the Novogene platform, and approximately 43 to 47 million high-quality reads (with a Fast QC quality score >38) were obtained for each biological replicate (Appendix A). Among all the reads obtained, 76 to 79% could be mapped to a unique chromosomal position. The expression level of each transcript in the sample was represented by fragments per kilobase of transcript per million fragments mapped (FPKM). A strong linear correlation was observed among all three replicates of the same sample, suggesting little differences among replicates (Appendix A). DEGs were identified by comparing the transcriptomes of *FaSnRK1α*-RNAi (fasnrk1a) and WT (control) roots. We identified a total of 6620 DEGs, among which 3086 and 3534 were upregulated and downregulated, respectively, in the *FaSnRK1α*-RNAi plants compared with the WT plants (Figure 5A). These results revealed that the expression of a large number of genes was altered and that these genes were either directly or indirectly regulated by SnRK1. Next, GO terms were assigned to the DEGs (Appendix A). A number of genes were involved in various cellular components, molecular functions, and biological processes (Figure 5B). Among the DEGs in biological processes category, defence response, response to biotic stimulus, and cellular carbohydrate metabolic process were the obviously enriched GO terms. Then, functional annotations of both upregulated and downregulated genes and Kyoto Encyclopaedia of Genes and Genomes (KEGG) pathway enrichment analysis were performed (Appendix A). Glycolysis/gluconeogenesis and pyruvate metabolism, which are related to respiration, were the main KEGG pathways that were regulated in the *FaSnRK1α*-RNAi roots (Figure 5C). Notably, the expression levels of 42 genes involved in glycolysis and 30 genes involved in pyruvate metabolism were significantly regulated in *FaSnRK1α*-RNAi roots (Appendix A; Figure 5D). These results provide evidence that *FaSnRK1α* affects root respiration and enhances resistance to anaerobic stress.

### 2.5. FaSnRK1α Regulates the Expression of Anoxia-Related Genes and ERFVIIs

Prior studies indicated a potential function of SnRK1 in the response to hypoxia [18]. In this study, *FaSnRK1α* was induced in the roots of WT plants in response to waterlogging (Figure 6A). To determine whether *FaSnRK1α* is essential for the expression of hypoxia-induced genes such as ADH and PDC [18], RT–qPCR was used to measure the expression of *FaADH1* and *FaPDC1*. As shown in Figure 6B,C, *FaADH1* and *FaPDC1* were upregulated in the WT in response to waterlogging and were further upregulated in the roots of the *FaSnRK1α*-OE plants. More importantly, gene induction in response to waterlogging was significantly downregulated in the roots of *FaSnRK1α*-RNAi plants. Thus, *FaSnRK1α* is important for the induction of *ADH1* and *PDC1* in response to waterlogging.

The ethylene pathway is the main hormone signalling pathway in plants in response to soil waterlogging and submergence. Among all the DEGs, we identified 47 ethylene-responsive transcription factor (*ERF*)-encoding genes that were regulated by *FaSnRK1α* (Appendix A). In Arabidopsis, five ERF VIIs, namely, *HRE1*, *HRE2*, *RAP2.2*, *RAP2.3* and *RAP2.12*, have been identified as key regulators of waterlogging and hypoxic tolerance [14,16]. Therefore, the homologous genes *FaHRE2*, *FaRAP2.3* and *FaRAP2.12* were identified and analysed in the roots of *FaSnRK1α*-OE and *FaSnRK1α*-RNAi strawberry plants. *FaHRE2* and *FaRAP2.12* were upregulated in response to waterlogging and were further upregulated in the roots of *FaSnRK1α*-OE plants and downregulated in the roots of *FaSnRK1α*-RNAi plants. *FaRAP2.3* was downregulated in response to waterlogging, but there was no obvious difference in the *FaRAP2.3* expression level between the roots of *FaSnRK1α*-OE plants and the roots of *FaSnRK1α*-RNAi plants (Figure 6D–F). Overall, these results indicate that *FaSnRK1α* enhances hypoxia tolerance by regulating the ethylene signalling pathway.

## 3. Discussion

As a common abiotic stress, waterlogging has a serious negative impact on plant growth and development. In the process of waterlogging stress, SnRK1 kinase is activated in response to energy starvation, after which the activated form then regulates a large number of defence-related genes and establishes defence mechanisms in plants. Although there is some evidence that SnRK1 can participate in resistance to anaerobic (flooded) conditions in rice and Arabidopsis [18,25,28], it is unclear how plants respond specifically to flooding stress.

As a key component of the cell signalling network, plant SnRK1 orchestrates the formation of transcriptional networks to promote catabolism and suppress anabolism and to maintain cellular energy homeostasis under stress conditions [29,30]. In this study, we showed that *FaSnRK1α* can increase root activity and respiration rates under waterlogging conditions (Figure 2) and decrease ROS accumulation (Figure 4). Furthermore, *FaSnRK1α* may regulate anaerobic respiration through the ethylene signalling pathway in response to waterlogging stress.

Under waterlogging stress, energy-related metabolism shifts from aerobic respiration to anaerobic respiration, and the ATP produced by anaerobic respiration per molecule of Glc is much lower than that by aerobic respiration. Therefore, a higher level of anaerobic respiration is the key to maintaining a sufficient energy supply for plants under flooding stress. In this study, we found that *FaSnRK1α* promoted the glycolysis pathway and the anaerobic metabolism of pyruvate by increasing the activities of glycolysis-related enzymes (HK, PFK and PK) and fermentation-related enzymes (PDC, ADH and LDH) under hypoxic stress (Figure 3). Transcriptomic analysis also revealed that 72 genes related to glycolysis and pyruvate metabolism were regulated by *FaSnRK1α* (Figure 5), which provided evidence that *FaSnRK1α* enhanced anaerobic respiration. We also showed that the expression levels of *FaADH1* and *FaPDC1* were upregulated under waterlogging stress and further upregulated in response to *FaSnRK1α* (Figure 6). This confirms results obtained in A. thaliana [18], where the overexpression of *KIN10* or *OsSnRK1* enhanced anaerobic respiration and increased the expression level of *ADH1* and *PDC1* under submergence. Therefore, we think that *FaSnRK1α* enhances anaerobic respiration by regulating the expression levels of *FaADH1* and *FaPDC1* and increasing the activity of anaerobic respiration-related enzymes. Alcoholic fermentation and lactic acid fermentation compensate for the energy deficit caused by hypoxia to some extent. However, fermentation products have certain toxic effects on cells, and there are still some limitations for plants to adapt to hypoxic stress through the regulation of anaerobic respiration and metabolism.

The imbalance of aerobic respiration and anaerobic metabolism induced by flooding stress increases the accumulation of ROS in plants. Excessive amounts of ROS can lead to oxidative damage to biological macromolecules. Several antioxidant enzymes, such as SOD, POD and CAT, play important roles in the process of scavenging ROS [31]. In our previous study, overexpression of *PpSnRK1α* was shown to increase the ROS scavenging capacity and the expression of antioxidant enzyme genes under salt stress [32]. In our current investigation, *FaSnRK1α* regulated ROS accumulation and antioxidant enzyme activities under waterlogging (Figure 4), suggesting that *FaSnRK1α* plays a positive role under hypoxic or anaerobic stress. ROS can also act as secondary messengers to mediate responses to waterlogging stress [33,34,35]. Some protein kinases can sense ROS signals and respond to stress through phosphorylation or dephosphorylation. However, whether *FaSnRK1α* responds to ROS signals and its underlying response mechanism need further exploration.

Ethylene is a key player in regulating the low-oxygen stress response in plants. ROS are important signals that can enhance ethylene signalling and *ADH1* expression during oxygen deprivation [36,37,38]. A previous study demonstrated an interaction between SnRK1.1 and ethylene signalling, and compared with WT fruit, *PpSnRK1α*-overexpressing fruit were shown to present greater ethylene release rates [39]. *HRE2* was shown to be expressed mostly in the roots of plants under hypoxic conditions [14]. Paul et al. [13] found that stabilized RAP2.12 increased the activities of fermentation-related enzymes and the accumulation of fermentation products. Upon hypoxia, RAP2.12 moves from the plasma membrane into the nucleus to specifically activate the expression of anoxia-related genes such as *PDC1* and *ADH1*, which are involved in ethanol fermentation [40]. In our study, two *ERFVIIs* (*FaHRE2* and *FaRAP2.12*) were upregulated in response to waterlogging and were further upregulated in the roots of *FaSnRK1α*-OE plants and downregulated in the roots of *FaSnRK1α*-RNAi plants (Figure 6). These results suggested that *FaSnRK1α* may be involved in ethylene signalling by regulating the expression of *FaHRE2* and *FaRAP2.13* under waterlogging.

## 4. Summary

In summary, the results of this study indicate that *FaSnRK1α* plays a role in modulating gene expression and enzyme activity related to glycolysis and fermentation during waterlogging stress. Furthermore, we showed that *FaSnRK1α* modulates antioxidant enzyme activity to reduce ROS accumulation. We also showed that, by regulating the expression of *ERFVIIs*, *FaHRE2* and *FaRAP2.13*, *FaSnRK1α* enhanced the tolerance of roots to hypoxia. However, the mechanism through which SnRK1 regulates *HRE2* and *RAP2.12* expression needs further investigation.

## 5. Materials and Methods

### 5.1. Vector Construction and Agrobacterium-Mediated Infiltration

For overexpression of *FaSnRK1α* (MN933920.1), the sequence was amplified and cloned into the expression vector pCAMBIA1302 by the homologous recombination method. Empty pCAMBIA1302 and pCAMBIA1302-*FaSnRK1α* vectors were subsequently transformed into *Agrobacterium rhizogenes* strain K599 (AC1080, Shanghai Weidi Biotechnology Co., Ltd., Shanghai, China)) following the instructions. For silencing of the *FaSnRK1α* gene, three silenced target fragments were selected from the http://plantgrn.noble.org/pssRNAit/ (accessed on 20 February 2020) website according to their coding DNA sequence (CDS) (Figure 7). The sequences *FaSnRK1α*-1, *FaSnRK1α*-2 and *FaSnRK1α*-3 were cloned into a TRV2 expression vector. TRV1 (the auxiliary vector of virus-induced gene silencing (VIGS) system); empty TRV2; or the TRV2-*FaSnRK1α*-1, TRV2-*FaSnRK1α*-2 and TRV2-*FaSnRK1α*-3 TRV2 derivative vectors were transformed into *A. rhizogenes* strain K599. All the amplification primers used are shown in Appendix A.

Healthy and uniform ‘Miaoxiang 7’ strawberry seedlings growing in our testing base (Tai’an, China) were selected for agroinfiltration using the following procedure. Single colonies were selected and cultured in 10 mL of TY media supplemented with antibiotics (100µg·mL^−1^ kanamycin and 50 µg·mL^−1^ streptomycin) overnight at 28 °C and then, when the optical density of K599 solution (OD600) was 0.6, the *A. rhizogenes* cells were collected by centrifugation and resuspended in 2-(N-morpholino)ethanesulfonic acid (MES) buffer solution (10 mmol·L^−1^ MES-KOH (pH 5.2), 10 mmol·L^−1^ MgCl_2_ and 100 µmol·L^−1^ acetosyringone). After 2 h of incubation in the dark, the suspension was applied to the strawberry roots for infection. As shown in Figure 8, most of the strawberry roots were cut off and the remaining roots were immersed in the bacterial suspension for 30 min. Then, the strawberry seedlings were grown in pots filled with a mixture of peat and vermiculite (1:1, *v*/*v*). At one month after injection, when the roots were well developed, the original roots were cut off, and gene expression was detected and measured using an in vivo imaging system and RT–qPCR, respectively.

### 5.2. Plant Materials and Treatment

For phenotypic assays, data were collected from 75 plants of five different genotypes in three independent experiments. After *A. rhizogenes* K599 infection, the wild-type (WT), FaSnRK1-OE (pCAMBIA1302-*FaSnRK1α*), OE-C (empty pCAMBIA1302), FaSnRK1-RNAi (TRV1 and TRV2-*FaSnRK1α*-2) and RNAi-C (TRV1 and empty TRV2) potted strawberry seedlings were grown in a growth chamber for 20 d (to allow recovery) and then placed into distilled water for waterlogging treatment. The root samples were collected at days 0, 3, 6, 9 and 12 after treatment, and all root samples were obtained from the organs that presented green fluorescent protein (GFP) signals via fluorescence imaging.

### 5.3. Determination of Root Activity and Root Respiration Rate

Root activity was measured by the triphenyltetrazolium chloride (TTC) method [41]. Approximately 1.0 g of fresh roots were washed with distilled water, immersed in 10 mL of mixed 0.4%TTC: phosphate buffer (pH 7.0) (*v*:*v* = 1:1) solution. After dark incubation for 4 h, 2 mL of 1 mol·L^−1^ H_2_SO_4_ was added to stop the reaction. The TTC reduction product triphenylformazan (TTF) was extracted with 95% ethanol. The optical density at 485 nm (OD485) was measured and the root activity was calculated according to the standard curve.

Root respiration rate was measured by Oxytherm type oxygen electrode of Hansateach (Hansha Scientific Instruments Limited, Tai’an, China) [41]. Approximately 0.1 g of white roots were selected and cut into small segments. The root segments were placed for about 15 min to eliminate the effect of injured respiration, and then were put into the cuvette (contains 1.5 mL distilled water) to measure the total respiration rate.

### 5.4. Determination of Respiration-Related Enzyme Activities

Approximately 0.5 g of frozen, finely ground tissue powder was homogenized in 2 mL of cold 50 mmol·L^−1^ Tris-HCl buffer solution (5 mmol·L^−1^ MgCl_2_, 5 mmol·L^−1^ β-mercaptoethanol, 1 mmol·L^−1^ ethylenediaminetetraacetic acid, 1 mmol·L^−1^ ethylene glycol tetraacetic acid, 0.1 mmol·L^−1^ phenylmethyl sulfonyl fluoride (pH 6.8)), after which it was transferred to a prechilled microfuge tube and clarified by centrifugation at 12,000× *g* for 5 min at 4 °C. Hexokinase (HK), phosphofructokinase (PFK), and pyruvate kinase (PK) activities were determined by enzyme-linked immunosorbent assay (ELISA) kits (MB-10607, MB-10759 and MB-4689, Jiangsu Meibiao Biotechnology Co., Ltd., Yancheng, China) according to the manufacturer’s protocols. For LDH activity determination, 2.5 mL of Tris-NaCl-NADH buffer, 0.15 mL of enzyme solution and 0.5 mL of Tris-NaCl-pyruvate were used, and for ADH activity determination 1.47 mL of 50 mmol·L^−1^ TES buffer, 1.44 mL of H_2_O, 50 µL of enzyme solution and 20 µL of 40% aldehyde were used. For PDC activity determination, 940 µL of 50 mmol·L^−1^ MES buffer solution (pH 6.8), 1 mL of H_2_O, 0.1 mL of enzyme solution and 50 µL of 10 mmol·L^−1^ pyruvate were used. The optical density at 340 nm (OD340) was measured, and the ADH, LDH, and PDC activities are shown as micromoles per minute per gram of fresh weight (FW).

### 5.5. ROS Accumulation and Antioxidant Capacity

The H_2_O_2_ content was measured using a reagent kit (ml076343, Mlbio, Shanghai, China) as the manufacturer’s protocol. The O^2−^ content was quantified by the hydroxylamine oxidation method [42]. For antioxidant capacity determination, the plant roots were ground and homogenized in phosphate-buffered solution (pH = 7.8) and after centrifugation, the enzymes were extracted from the supernatant [43]. SOD activity was determined using the nitroblue tetrazolium (NBT) method, and 50% activity of inhibiting photochemical reduction of NBT was defined as one enzyme activity unit (U) [44]. CAT activity was determined using the ultraviolet absorption method, and a change in the optical density at 240 nm (OD240) of 0.1 within 1 min was defined as 1 U [45]. POD activity was determined using the guaiacol method, and an OD470 of 0.1 per minute was defined as 1 U [46]. SOD, POD, and CAT activities were shown as U·min^−1^ g^−1^ (FW).

### 5.6. RNA Isolation and Library Preparation

After Agrobacterium-mediated infiltration, the roots of WT and *FaSnRK1α*-RNAi of strawberry plants grown in substrate for 15 d were collected. The total amount and integrity of RNA were assessed using an RNA Nano 6000 Assay Kit of a Bioanalyzer 2100 system (Agilent Technologies, CA, USA). Library preparation, clustering and sequencing were performed by Beijing Novogene Co., Ltd. (Beijing, China).

### 5.7. RNA Sequencing and Differentially Expressed Gene (DEG) Analysis

The image data measured by the high-throughput sequencer were converted into sequence data (reads) by CASAVA base recognition, and all downstream analyses were based on clean high-quality data.

Reference genome and gene model annotation files were downloaded from the genome website directly. The index of the reference genome was constructed using HISAT2 (v2.0.5) and paired-end clean reads were aligned to the reference genome using HISAT2 (v2.0.5).

Analysis of the genes differentially expressed between wide type (control) and *FaSnR**K1α*-RNAi (fasnrk1a) was performed using the DESeq2 R package (v1.20.0). The resulting *p* values were adjusted using Benjamini and Hochberg’s approach for controlling the false discovery rate; padj ≤ 0.05 and |log2(fold-change)| ≥ 1 were set as the thresholds for significantly different expression.

Gene Ontology (GO) enrichment analysis of DEGs was implemented by the clusterProfiler R package (v3.8.1), for which gene length bias was corrected.

### 5.8. RT-qPCR

Total RNA was extracted from evenly mixed samples (1 g, the sample is a mixture of each separately treated sample) using a Fast Pure Plant Total RNA Isolation Kit (RC401, Vazyme, Nanjing, China) and reverse transcribed to generate the first strand cDNA using HiScript III RT SuperMix for qPCR (+gDNA wiper) (R323-01, Vazyme, Nanjing, China) according to the manufacturer’s instructions. Quantitative real-time PCR (RT-qPCR) consisted of three biological and three technical replicates and was carried out using the Bole CFX96 system (Bio-Rad, Hercules, CA, USA) and Ultra SYBR mixture (CW2601M, CWBIO, Beijing, China,). The calculation method for RT–qPCR was 2^−ΔΔCt^ with *FaACTIN* as the internal control. The specific primers used are listed in Appendix A.

## Figures and Tables

**Figure 1 ijms-23-04914-f001:**
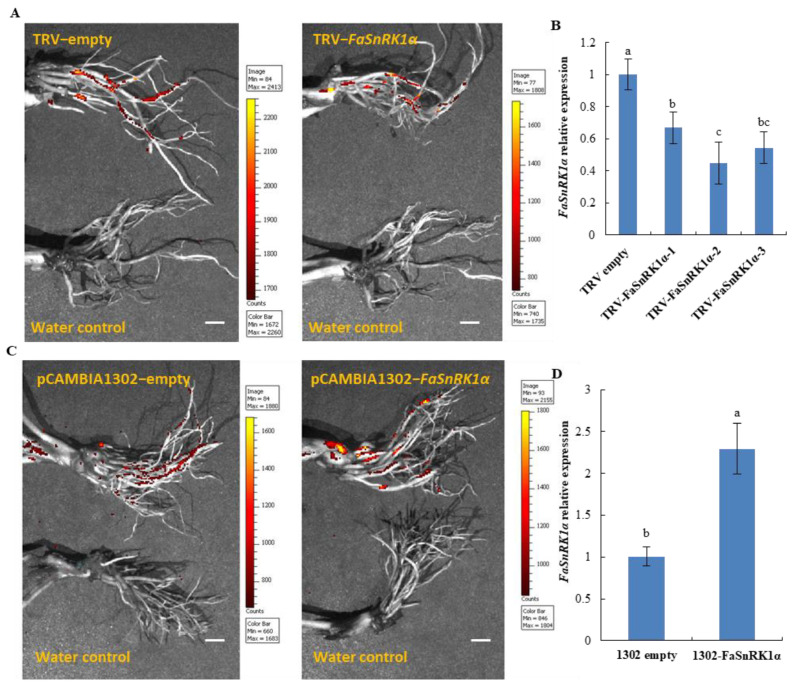
Generation and identification of roots in which *FaSnRK1α* is overexpressed or silenced. (**A**,**C**) Root fluorescence images of TRV-*FaSnRK1α* (RNAi) and pCAMBIA1302-*FaSnRK1α* (OE) transgenic roots one month after infection with *A. rhizogenes* strain K599. The fluorescence intensity was recorded as photons per second per square centimetre. The signal intensity scale is given on the right side of the image, and the colour of the signal represents the amount of the fluorescent protein present. Scale bar = 1 cm (**B**,**D**) Expression of the *FaSnRK1α* gene in the roots. All the root samples were obtained from the parts that exhibited GFP signals by fluorescence imaging. The transcript levels were measured by RT–qPCR with specific primers. *FaACTIN* was the internal control. The expression levels of indicated genes in the roots transferred into empty vectors were normalized to 1. The error bars represent the SDs of three biological replicates. The different lowercase letters indicate significant differences at *p* < 0.05.

**Figure 2 ijms-23-04914-f002:**
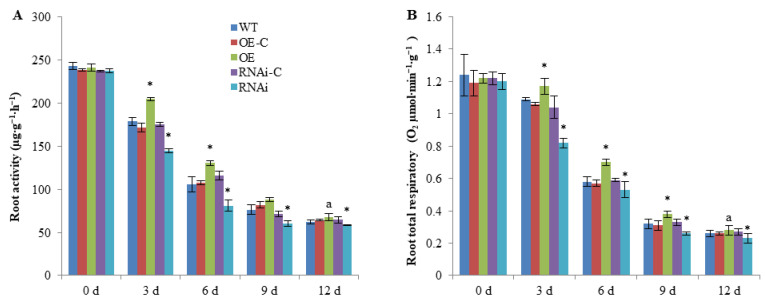
Root activity (the ability of roots to absorb water or mineral elements) (**A**) and root total respiration rate (**B**) in WT, *FaSnRK1α* overexpression (OE) and *FaSnRK1α*-silenced (RNAi) lines after waterlogging treatment. WT: Wild-type roots; OE-C and RNAi-C denote controls for the *FaSnRK1α*-OE or *FaSnRK1α*-RNAi roots (i.e., their roots were transformed with corresponding empty vectors). All the root samples were obtained from the parts that exhibited GFP signals by fluorescence imaging. The data are shown as the means ± SDs of three independent biological replicates. An * (*p* < 0.05) above the bar indicates a statistically significant difference from the corresponding control as determined by Student’s *t*-test. The lowercase letter indicates significant differences at *p* < 0.05.

**Figure 3 ijms-23-04914-f003:**
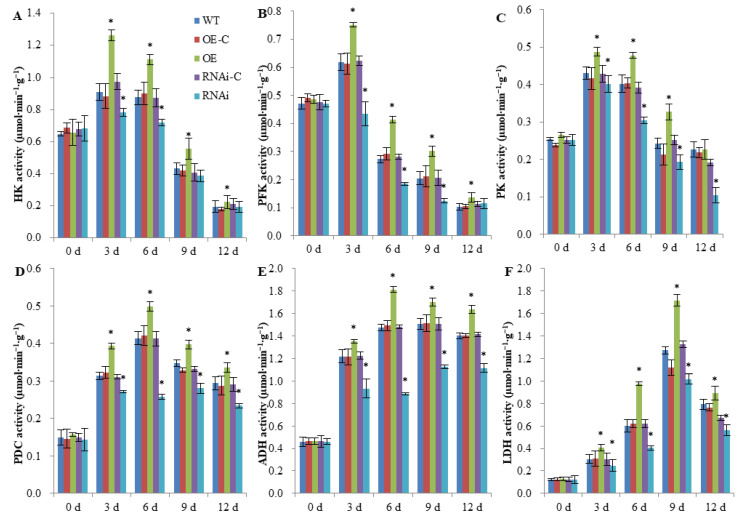
Activities of enzymes related to anaerobic respiration metabolism in WT, OE and RNAi roots after waterlogging. (**A**–**C**) Activity of key enzymes in the glycolysis pathway. (**D**–**F**) Activities of related enzymes during alcohol and lactate fermentation. HK: hexokinase; PFK: phosphofructokinase; PK: pyruvate kinase; PDC: pyruvate decarboxylase; ADH: alcohol dehydrogenase; LDH: lactate dehydrogenase. All the root samples were obtained from the parts that presented GFP signals by fluorescence imaging. The data are shown as the means ± SDs of three independent biological replicates. An * (*p* < 0.05) above the bar indicates a statistically significant difference from the corresponding control as determined by Student’s *t*-test.

**Figure 4 ijms-23-04914-f004:**
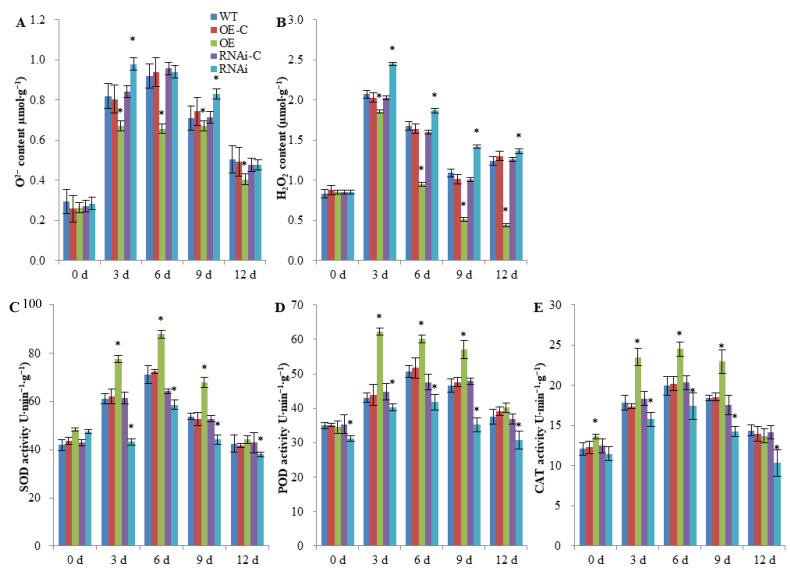
ROS accumulation and antioxidant enzyme activities in FaSnRK1α overexpression (OE) and FaSnRK1α-silenced (RNAi) roots. (**A**,**B**) O^2−^ (**A**) and H_2_O_2_ (**B**) contents in the WT, OE and RNAi roots after waterlogging. (**C**–**E**) SOD (**C**), POD (**D**) and CAT (**E**) activities in the WT, OE, and RNAi roots after waterlogging. All the root samples were obtained from the parts that presented GFP signals by fluorescence imaging. The data are shown as the means ± SDs of three independent biological replicates. An * (*p* < 0.05) above the bar indicates a statistically significant difference from the corresponding control as determined by Student’s *t*-test.

**Figure 5 ijms-23-04914-f005:**
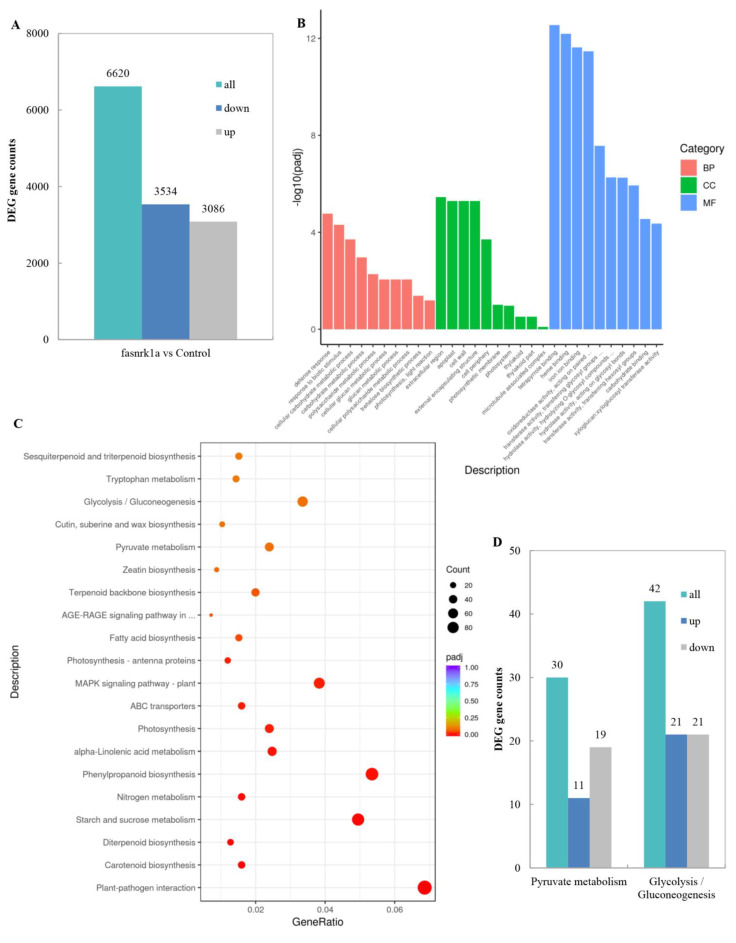
Identification and functional annotation of DEGs. (**A**) DEG identification. (**B**) GO classification analysis of the DEGs. (**C**) KEGG classification analysis of the DEGs. (**D**) DEGs involved in the glycolysis and pyruvate metabolic pathways.

**Figure 6 ijms-23-04914-f006:**
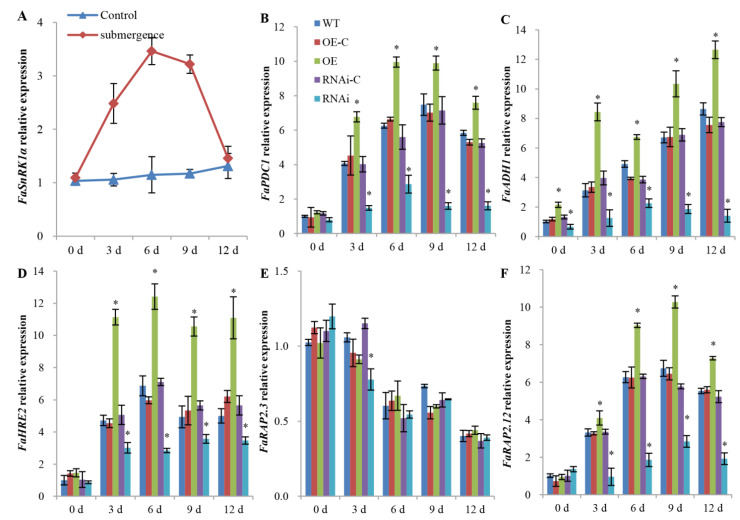
Expression profiles of *FaSnRK1α* (**A**), anoxia-related genes (**B**,**C**) and group VII *ERFs* (**D**–**F**). (**A**) Changes in *FaSnRK1α* expression levels in WT roots after waterlogging. (**B**,**C**) Effects of waterlogging on the expression of hypoxia-induced genes in the roots of *FaSnRK1α*-OE and the roots of *FaSnRK1α*-RNAi plants. The relative expression levels of *FaPDC1* (AF193791.1) and *FaADH1* (XM004290519.2) were measured. PDC: pyruvate decarboxylase; ADH: alcohol dehydrogenase. (**D**–**F**) Effects of waterlogging on the expression of *ERFVIIs* in the roots of *FaSnRK1α*-OE plants and the roots of *FaSnRK1α*-RNAi plants. The relative expression levels of *FaHRE2* (MH332957.1), *FaRAP2.3* (MH332956.1), and *FaRAP2.12* (MH332958.1) were measured. The transcript levels were detected via RT–qPCR with specific primers. *FaACTIN* was the internal control. The expression levels of indicated genes in WT roots under normal conditions (0 d) were normalized to 1. The error bars represent the SDs of three biological replicates. An * (*p* < 0.05) above the bar indicates a statistically significant difference from the corresponding control as determined by Student’s *t*-test.

**Figure 7 ijms-23-04914-f007:**
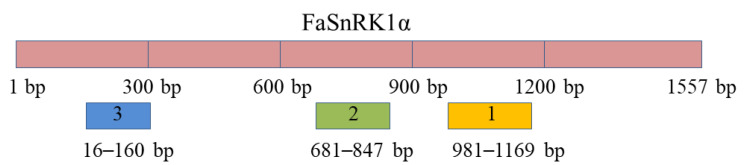
Length and location of three *FaSnRK1α* silencing targets.

**Figure 8 ijms-23-04914-f008:**
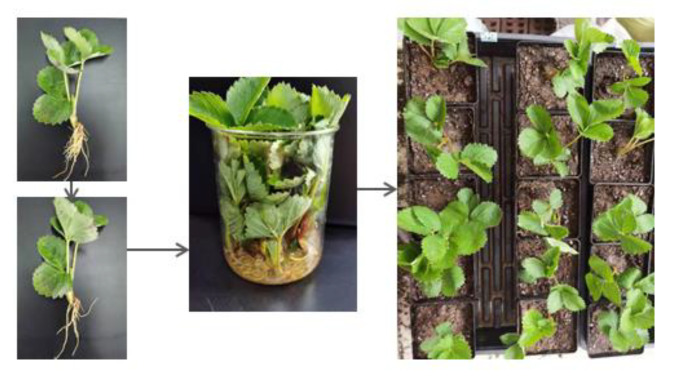
Schematic diagram of *A. rhizogenes* K599 infection.

## Data Availability

The data that support the findings of this study are available from the corresponding author upon reasonable request. All relevant data can be found within the manuscript and its Appendix A.

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
