# Peer review of "Strawberry FaSnRK1α Regulates Anaerobic Respiratory Metabolism under Waterlogging"

_ijms, 2022, doi:10.3390/ijms23094914_

Round 1
Reviewer 1 Report
Strawberry FaSnRK1α regulates anaerobic respiratory metabolism under waterlogging
The work presented by Luo and co-authors deals with a study carried out in strawberry aiming to investigate the role of the gene FaSnRK1α in plant response upon waterlogging. The authors follow different approaches, investigating transcriptional changes by RNA-seq and RT-qPCR, enzyme activity, and ROS measurement. Several analyses at the root physiological level were also carried out. The topic of the manuscript is interesting and reaches the scope of the IJMS, however, important flags were detected which make not possible to accept it in its present form. Important informations are missing, which includes some methods. The RT-qPCR analysis is based on the use of a single reference gene without demonstration of its applicability. The publication of the RNA-seq data was not done, the statistical analysis is poorly explained, and it was also not possible to access the supplementary material.
I suggest authors to improve the manuscript considering the comments I am including below.
Specific comments:
Line 99: the authors should add as supplementary materials the images of GPR control.
Line 107: “was 2.3 times” higher or lower?
The authors start referring to Fig. 1C instead to start with Fig. 1A. Fig. 1A, 1B, and 1D are not referred to within the text. Please confirm that all figures are referred into the text.
Fig. 1A) indicates the length and location of the three silencing targets but no further details are given. Why have the authors selected those regions? Are those regions responsible for an important region in the functionality of the enzyme?
Fig. 2 and Fig. 3 present a different representation of significant differences. I suggest uniformizing that. Authors must include the procedure followed for statistical analysis.
Line 213: “Three biological replicates of the WT and three transgenic lines were sequenced” which is not correct. The transcriptome was sequenced and that sentence can led readers to think about genome sequencing.
Fig. 5 shows the number of genes linked to pyruvate metabolism and glycolysis but failed in showing the expression pattern. It will be interesting to have an idea if all genes show a similar level of expression. Additionally, it will be interesting to have an idea about the involvement of alternative transcripts for each gene. It is not clear how the authors made the identification of the different genes, have they considered the alternative transcripts as a single gene?
References are not uniformized, please see lines 244 (Baena-Gonzalez et al., 2007; Cho et al., 2016) and 258 [14, 16].
Line 370: “following the instructions” provided by whom?
Line 374: It is not clear the use of TRV1.
Line 376: include A. rhizogenes strain K599.
368: It is not clear the aim of using pCAMBIA1300 and pCAMBIA1302.
379: “supplemented with antibiotics” the information regarding the antibiotics must be detailed.
Line 400: “presented had green fluorescent protein” remove had.
Line 401: please explain the meaning of “root activity” and add the procedure followed for those measurements and also for the oxygen electrode method. The number of biological replicates considered and measurements per biological replicate must be added.
Line 447: “between the two groups” the groups must be identified.
Information regarding the availability of RNAseq data in public databases is not given, which doesn’t allow reviewers to confirm results.
Line 455: please revise the text considering the biological replicates.
Regarding the transcript analysis, the authors have not followed the MIQE guidelines for RT-qPCR (Bustin et al., 2009). Accordingly, it must be used RT-qPCR (reverse transcription quantitative real-time polymerase chain reaction) instead of qRT-PCR. All the technical aspects of RT-qPCR experiments should fit the paper of Bustin et al. (2009).
The method selected for the normalization of the target genes, the 2-ΔΔCt method, is not the most appropriate. The efficiency is not considered by that method, and it requires the use of a calibrator sample, which was not described in the manuscript.
Additionally, a single gene was used as reference without considering the demonstration of its applicability as reference gene under the experimental conditions. The applicability of a selected gene for data normalization depends on a diversity of factors, including environmental stresses. In the present research, the authors have not correctly stated the applicability of that gene. The use of a single gene as reference for data normalization is also not acceptable, at least a minimum of two genes should be considered.
No information is provided regarding technical replicates, primers specificity, efficiency, and identification of putative contaminations with DNA in cDNA samples. Statistical analysis was not described.
Reviewer 2 Report
The manuscript “Strawberry FaSnRK1α regulates anaerobic respiratory metabolism under waterlogging” by Luo, et al., is based on the investigation of the regulation of respiratory rate and tolerance to waterlogging in strawberry plants. The research objectives were achieved by using systematic approaches for the estimation of waterlogging stress. It is valuable in the horticulture sector to understand the plant behavior under waterlogging stress conditions.
The language used in the manuscript is not up to the standard for publication, several major and minor mistakes are shown, and many of the phrases are not clear. It should go through complete proofreading by a native speaker.
Title: The title is good and reflects the work done by the authors.
Abstract: Abstract summarized the work precisely.
Keywords should be different from the title words.
Introduction
The introduction is technically fine and to the point, but there is a need to add information regarding the scope and agronomic importance of the strawberry crop. The introduction of the manuscript is quite lengthy, which is a good thing but on the other hand, it makes it look too exaggerated as well. The authors should make it concise and to the point because there are many irrelevant bibliographies in this section as well.
Moreover, at the end of this section, the authors should also mention the future prospects of this experiment and how it can benefit the researchers/stakeholders in this sector.
Materials and Methods
The methodology of the proposed study is fine and well-designed.
Results and Discussion
The other critical weakness of this manuscript is the Discussion part. It is not comprehensively described. In the discussion, some conclusions are too exaggerated. Authors should focus on specific and to the point information relating to their experiment and elaborate on their achievements to justify their arguments developed in the results data. This section needs critical revisions
Figure 1 B & D needs scale bars.
The conclusions section should also be added separately, and it should be a little more elaborative.
To sum up, the manuscript can find interest among specialists when the comments will be considered.
Good Luck!
